# Hydrothermal and Entropy Investigation of Nanofluid Natural Convection in a Lid-Driven Cavity Concentric with an Elliptical Cavity with a Wavy Boundary Heated from Below

**DOI:** 10.3390/nano12091392

**Published:** 2022-04-19

**Authors:** Aiman Alshare, Aissa Abderrahmane, Kamel Guedri, Obai Younis, Muhammed Fayz-Al-Asad, Hafiz Muhammed Ali, Wael Al-Kouz

**Affiliations:** 1Mechanical and Maintenance Engineering, German Jordanian University, P.O. Box 35247, Amman 11180, Jordan; 2Laboratoire de Physique Quantique de la Matière et Modélisation Mathématique (LPQ3M), University of Mustapha Stambouli of Mascara, Mascara 29000, Algeria; a.aissa@univ-mascara.dz; 3Mechanical Engineering Department, College of Engineering and Islamic Architecture, Umm Al-Qura University, P.O. Box 715, Makkah 21955, Saudi Arabia; kmguedri@uqu.edu.sa; 4Department of Mechanical Engineering, College of Engineering at Wadi Addawaser, Prince Sattam Bin Abdulaziz University, Al-Kharj 16278, Saudi Arabia; oubeytaha@hotmail.com; 5Department of Mathematics, Bangladesh University of Engineering and Technology, Dhaka 1000, Bangladesh; fayzmath.buet@gmail.com; 6Mechanical Engineering Department, King Fahd University of Petroleum and Minerals, Dhahran 31261, Saudi Arabia; hafiz.ali@kfupm.edu.sa; 7Interdisciplinary Research Center for Renewable Energy and Power Systems (IRC-REPS), King Fahd University of Petroleum and Minerals, Dhahran 31261, Saudi Arabia; 8College of Engineering and Technology, American University of the Middle East, Kuwait 15453, Kuwait; wael.kouz@aum.edu.kw

**Keywords:** mixed convection, nanofluid, entropy generation, lid-driven cavity

## Abstract

This work investigates mixed convection in a lid-driven cavity. This cavity is filled with nanofluid and subjected to a magnetic field. The concentric ovoid cavity orientation (γ),  0–90°, and undulation number (N), 1–4, are considered. The Richardson number (Ri) varies between 1 and 100. The nanofluid volume fraction (φ) ranges between 0 and 0.08%. The effect of the parameters on flow, thermal transport, and entropy generation is illustrated by the stream function, isotherms, and isentropic contours. Heat transfer is augmented and the Nusselt number rises with higher Ri, γ, N, and φ. The simulations show that the heat transfer is responsible for entropy generation, while frictional and magnetic effects are marginal.

## 1. Introduction

Combined convective flowing and temperature distribution into a wavy frame enclosure imposed by nanofluids have an immeasurable real-life enrollment in diverse industrial systems, engineering, and domestic processes such as heat exchangers, nuclear reactor technologies, solar collectors, electric equipment, refrigeration units, and others. Choi [1] reviewed nanofluid technology for current and future research. Individual sets of geometries, designs, and types of fluid, scrutinized in the recent year, incorporated wavy [2,3,4,5], square [6,7], triangular [8,9], rectangular [10,11], and hexagonal shapes [12], and so on. Mixed convection passing inside a wavy triangular box, loaded by nanofluid exerting viscosity criteria, was explored by Nasrin et al. [13]. Fares et al. [14] considered magneto-free convective inside a non-Darcy porous enclosure, using hybrid nanofluid enclosing an adiabatic rotating cylinder. Srinivas Acharya and Kumar [15] investigated mixed convection across an oblique wavy surface in a non-Darcy porous media saturated with nanofluids in the presence of radiation. Fares et al. [16] investigated the effect of a magnetic field on combined convection in a vented cavity using nanofluid flow. Ahmed and Aly [17] studied combined convection in a sloshing porous chamber filled with nanofluid and an internal heat source. Slimani et al. [18] investigated the spontaneous convection flow of Al_2_O_3_-Cu/water nanofluid in a conical form enclosure regulated by a magnetic field. Nada and Chamkha [19] studied the convection of a nanofluid in a driven enclosure surrounded by a wavy frame wall. Misirlioglu et al. [20] investigated natural convection numerically inside an oblique wavy cage. Mushate [21] studied the computational fluid dynamics simulation of natural convection in a wavy cavity filled with a porous substance. 

Additionally, the results indicated that the rate of heat evacuation increases as the Rayleigh number increases and decreases as the amplitude increases. Sheremet and Pop [22] explored natural convection in a wavy hollow filled with a nanofluid and surrounded by sinusoidal heat distributions on both level sidewalls. Shenoy et al. [23] investigated convective flow and heat removal from wavy coverings. The writers of this book have extensive experience in basic convection, heat transmission in wavy frames, saturated viscous fluids, and nanofluids porous media. Cheong et al. [24] investigated natural convective heating and heat production in a wavy enclosure. Recently, Asad et al. [25] investigated the heat transport properties of an internal chamber with unrestricted convection flow and vertical wavy frame walls. Zahan et al. [26] investigated the effect of (magneto hydro-dynamics) MHD on conjugate heat variation in a rectangular nanofluid container. Recent research on a nanofluid and wavy cavity, and the impact of a magnetic field, is accessible [27,28,29,30,31].

Additionally, because of the inquiry of flow construction and heat replacement within the enclosure with a fin-like radiator in cars, computer CPU heat sinks, power plant heat exchangers, and heat shifting devices have various technical applications. Sun et al. [32] investigated numerically mixed convection using conductive triangular fins in lid-driven enclosures. They reported that the triangular fin is an appropriate control parameter for the flow structure and heat transfer rate. Elatar et al. [33] used natural laminar convection to generate heat within a square frame with an adiabatic horizontal wall with a unique horizontal fin at various places and lengths connected to the heated wall. They examined the effect of fin placement and frame length on flow construction and heat removal components. Palaniappan et al. [34] investigated the impact of parallel insulated baffles inside open enclosures. By examining the literature records, it was discovered that further actions pertinent to the current research might be located in [35,36,37,38,39,40,41,42,43,44,45,46,47,48,49].

This study investigates the influence of various factors on the heat transfer characteristics of stable mixed convection in a lid-driven inclined square cavity, including the effect of varying volume fractions of the nanofluid, MHD, boundary annulations, and obstruction direction. The governing equations are solved using the Galerkin finite element method (GFEM) [7]. The acquired data are shown graphically using isotherms, streamlines, Bejan, and Nusselt values.

## 2. Physical Model

Depicted in Figure 1 is a lid-driven square cavity (LDSC) that is isothermal (T_H_), heated from below, with the top surface maintained at a uniform cold (Tc) temperature. In contrast, the lateral surfaces and the concentric oval-shaped cavity are adiabatic. The LDSC contains a nanofluid Al_2_O_3_, dispersed uniformly. The wavy boundary is sketched according to Equation (1), where A = 0.04, and the undulation, N, varies from 1 to 4.

### The Governing Equations

The governing equations (Equations (2)–(4)) are the continuity, momentum, and energy equations [50] for laminar mixed convection and are as follows: (1)$$\frac{\partial u}{\partial \;x}+\frac{\partial v}{\partial \;y}=0$$
(2)$$\rho_{nf}\left(u\frac{\partial \;u}{\partial \;x}+v\frac{\partial u}{\partial \;y}\right)=-\frac{\partial \;P}{\partial \;x}+\mu_{nf}\left(\frac{\partial^{2}u}{\partial \;x^{2}}+\frac{\partial^{2}u}{\partial \;y^{2}}\right)$$
(3)$$\rho_{nf}\left(u\frac{\partial \;v}{\partial \;x}+v\frac{\partial v}{\partial \;y}\right)=-\frac{\partial \;P}{\partial \;y}+\mu_{nf}\left(\frac{\partial^{2}v}{\partial \;x^{2}}+\frac{\partial^{2}v}{\partial \;y^{2}}\right)\;+\rho_{nf}\beta_{nf}g(T-T_{c})\;-\;\sigma_{nf}B^{2}v$$
(4)$${\left(\rho C_{p}\right)}_{nf}\left(u\frac{\partial \;T}{\partial \;x}+v\frac{\partial T}{\partial \;y}\right)=k_{nf}\left(\frac{\partial^{2}T}{\partial \;x^{2}}+\frac{\partial^{2}T}{\partial \;y^{2}}\right)$$

The thermophysical properties used in this study are the following Equations (5)–(10) [51] that use the tabulated values provided in Table 1:(5)$$\begin{array}{c} \mathrm{Density}\;\rho_{nf}=\left(1-\varphi \right)\rho_{f}+\varphi \rho_{P} \end{array}$$
(6)$$\begin{array}{c} \mathrm{Heat}\;\mathrm{capacity}\;{\left(\rho C_{p}\right)}_{nf}=\left(1-\varphi \right){\left(\rho C_{p}\right)}_{bf}+\varphi {\left(\rho C_{p}\right)}_{P} \end{array}$$
(7)$$\begin{array}{c} \mathrm{Thermal}\;\mathrm{expansion}\;\mathrm{coefficient}\;{\left(\rho \beta \right)}_{nf}=\left(1-\varphi \right){\left(\rho \beta \right)}_{f}+\varphi {\left(\rho \beta \right)}_{P} \end{array}$$
(8)$$\begin{array}{c} \mathrm{Electrical}\;\mathrm{conductivity}\;{\left(\sigma \right)}_{nf}=\left(1-\varphi \right){\left(\sigma \right)}_{f}+\varphi {\left(\sigma \right)}_{P} \end{array}$$
(9)$$\begin{array}{c} \mathrm{Thermal}\;\mathrm{conductivity}\;k_{nf}=k_{bf}\left(4.97\;\varphi^{2}+2.72\;\varphi +1\right) \end{array}$$
(10)$$\begin{array}{c} \mathrm{Dynamic}\;\mathrm{viscosity}\;\mu_{nf}=\mu_{bf}\left(123\;\varphi^{2}+7.3\;\varphi +1\right) \end{array}$$

Dimensionalization of the governing equation is carried out by scaling the equations using the characteristic length scale of the square cavity, *L*, and the velocity of the driven Lid, *U*_0_, as follows:(11)$$X=\frac{x}{L},\;Y=\frac{y}{L},\;U=\frac{u}{U_{0}},\;V=\frac{v}{U_{0}}\;\theta =\frac{T-T_{C}}{T_{h}-T_{C}},\;\theta_{s}=\frac{T_{s}-T_{C}}{T_{h}-T_{C}}\;P=\frac{p}{\rho_{nf}U_{0}^{2}},\;Pr=\frac{v_{f}}{a_{f}}$$

This then yields the dimensionless governing equations in [53,54,55,56]: where *Ri* = *Gr*/*Re*^2^ shows the Richardson number.

The dimensionless boundary conditions regarding Equations (12)–(15) are:

On the bottom wavy heated surface:(12)U=V=0, θ=1, 0≤X≤ 1, Y=0

On the top moving cold surface:(13)U=1, V=0, θ=0, 0≤X≤ 1, Y=1

On the left and the right surfaces:(14)$$U=V=0,\;\;\;\frac{\partial \theta }{\partial X}=0$$

At the concentric surface,
(15)$$U=V=0,\;\;\;\frac{\partial \theta }{\partial n}=0$$

The local Nusselt number evaluated at the heated bottom surface is defined by:(16)$$Nu_{s}=-\frac{k_{nf}}{k_{f}}{\left(\frac{\partial \theta }{\partial Y}\right)}_{Y=0}$$

The average Nusselt number evaluated at the heated part is:(17)$${\bar{Nu}}_{nf}=\int_{0}^{1}Nu_{s}dY$$

The entropy generation relation is given by [29]:(18)$$S=\frac{k_{nf}}{T_{0}^{2}}\left[{\left(\frac{\partial T}{\partial x}\right)}^{2}+{\left(\frac{\partial T}{\partial y}\right)}^{2}\right]+\frac{\mu_{nf}}{T_{0}}\left[\begin{array}{c} 2\left({\left(\frac{\partial u}{\partial x}\right)}^{2}+{\left(\frac{\partial v}{\partial y}\right)}^{2}\right){\;}^{}+{\left(\frac{\partial u}{\partial x}+\frac{\partial v}{\partial x}\right)}^{2} \end{array}\right]+\frac{\sigma_{nf}B^{2}V^{2}}{T_{o}}$$

In dimensionless form, local entropy generation can be expressed as:(19)$$S_{GEN}=\frac{k_{nf}}{k_{f}}\left[{\left(\frac{\partial \theta }{\partial X}\right)}^{2}+{\left(\frac{\partial \theta }{\partial Y}\right)}^{2}\right]+\frac{\mu_{nf}}{\mu_{f}}N_{\mu }\left[2\left({\left(\frac{\partial U}{\partial X}\right)}^{2}+{\left(\frac{\partial V}{\partial Y}\right)}^{2}\right)+{\left(\frac{\partial^{2}U}{\partial Y^{2}}+\frac{\partial^{2}V}{\partial X^{2}}\right)}^{2}\right]+N_{\mu }\frac{\sigma_{nf}}{\sigma_{f}}\;Ha^{2}V^{2}$$
where
$$N_{\mu }=\frac{\mu_{f}T_{0}}{k_{f}}{\left(\frac{\alpha_{f}}{L\left(\Delta T\right)}\right)}^{2}$$
is the irreversibility distribution ratio and $S_{GEN}=S_{gen}\frac{T_{0}^{2}L^{2}}{k_{f}{\left(\Delta T\right)}^{2}}.$ The terms of Equation (19) can be separated into the following form:(20)$$S_{GEN}=S_{\theta }+S_{\psi }+S_{B}$$
where $S_{\theta },\;S_{\psi ,}\;\mathrm{and}\;S_{B}$ are the entropy generation due to heat transfer irreversibility (HTI), fluid friction irreversibility (FFI), and entropy generation due to magnetic field effect, respectively:(21)$$S_{\theta }=\frac{k_{nf}}{k_{f}}\left[{\left(\frac{\partial \theta }{\partial X}\right)}^{2}+{\left(\frac{\partial \theta }{\partial Y}\right)}^{2}\right]$$
(22)$$S_{\psi }=\frac{\mu_{nf}}{\mu_{f}}N_{\mu }\left[2\left({\left(\frac{\partial U}{\partial X}\right)}^{2}+{\left(\frac{\partial V}{\partial Y}\right)}^{2}\right)+{\left(\frac{\partial^{2}U}{\partial Y^{2}}+\frac{\partial^{2}V}{\partial X^{2}}\right)}^{2}\right]$$
(23)$$S_{B}=N_{\mu }\frac{\sigma_{nf}}{\sigma_{f}}\;Ha^{2}V^{2}$$

It is appropriate to mention the Bejan number to determine the dominant, heat transfer, or fluid friction irreversibility. The Bejan number is defined as:(24)$$B_{e}=\frac{\int^{\;}S_{\theta }\mathrm{dXdY}}{\int^{\;}S_{GEN}\mathrm{dXdY}}$$

## 3. Numerical Method and Validation

The dimensionless controlling Equations (12)–(15) are augmented by the boundary conditions. The numerical solutions to Equations (18)–(22) are obtained using the Galerkin weighted residual finite element technique [53,54,55]. The issuing domain is discretized using non-uniform triangular components. The finite element equations are constructed using triangular elements with six nodes. The Galerkin weighted residual approach converts the non-linear partial differential equations to a system of integral equations. Each term is then solved using Gauss’s quadrature method. The objective is to obtain a set of non-linear algebraic equations that satisfy the boundary conditions. Grid-independent solutions are discovered by experimenting with a few grids. To conduct all simulations, a grid of 695,244 elements was used. To guarantee the numerical approach converted approved code is accurate, the velocity profile within the cavity with obstacles is compared with Iwatsu et al.’s work [56]. In Figure 2, a perfect agreement between the two results is obtained. 

## 4. Results and Discussion

To understand the salient features of this problem, the fluid motion streamlines, the temperature field isotherm contours, and entropy distribution depicted by the isentropic contours are utilized. The controlling geometric parameters are the orientation of the concentric ovoid cavity and the undulation number on the heated boundary. Heating from below gives rise to a buoyancy force in conjunction with the sliding top lid, which assists the fluid motion. Here, mixed convection and free convection regimes are investigated. The characterizing parameter is the Richardson number. The Hartmann number characterizes the intensity of the applied magnetic field. The thermo-physical controlling parameter is the nanoparticle volume fraction. The numerical experiments are conducted for Richardson number values (*Ri*: 1, 10, 50, 100), Hartman number values (*Ha*: 1, 25, 50, 100), orientation (γ: 0°, 30°, 50°, 90°), undulation (*N*: 1, 2, 3, 4), and nanoparticle volume fraction (φ: 0, 0.02, 0.04, 0.08). Figure 3a shows the effect of the Richardson number.

It is interesting to consider Δψ, as the difference between the maximum and minimum value of the stream function, as Ri is varied. We note that Δψ increases monotonically with Ri. There are two vortices at low Ri: a local vortex near the top moving lid and a global vortex with its sudo-center the oval-shaped cavity. At higher Ri, the local vortex is stretched vertically, and, eventually, the local vortex is merged into the global vortex. The influence of natural convection driven by buoyancy forces dominates the inertial forces of forced convection. Considering Figure 3b, more significant distortion of the isotherms is observed at a higher Ri number, meaning greater homogenization of the temperature field.

Furthermore, at higher Ri, the thermal boundary layer becomes thinner. The isotherms become condensed near the heated wavy boundary, implying enhanced heat transfer dominated by natural convection in contrast to the sparse thermal lines at low Ri, indicating a thick thermal boundary layer and less efficient thermal transport dominated by mixed convection. The merging and increased strength of the re-circulating fluid are reflected in the increase in frictional irreversibility, as depicted in the entropy generation plots shown in Figure 3c, near the wavy boundary and the moving boundary.

Shown in Figure 4 are the plots for the fixed Ri of 1 and the varying angles of 0°, 30°, 60°, and 90°. The effect of rotating the principal axes of the internal oval cavity from 0° to 90° is illustrated.

The tip of the major axis squeezes the effective area of the local vortex with an increasing ellipse major axes angle, shown in Figure 4a. The streamlining of the obstruction with the gravity axes invigorates the mixed convective heat transfer, as evident by the distortion in the isotherm plots, shown in Figure 4b, which leads to an increase in entropy generation, as shown in Figure 4c. At low Ri, a horizontally oriented cavity impedes the inertially driven flow from mixing. This is confirmed by Figure 5a which shows the average Nusselt number versus Richardson number at various angles.

At low Ri number, we note a monotonic and uniform increase in the average dimensionless heat transfer coefficient associated with incrementing the angle of the ovoid cavity from 0° to 90°. However, a marginal increase in the heat transfer coefficient is found at higher Ri numbers. Bejan number which represents the ratio of heat transfer to fluid friction irreversibility is illustrated in Figure 5b. Generally, there is a small increase in entropy as the angle of the ovoid cavity is increased which can be attributed to the distortion of the local vortex and nudge in the ratio entropy generated due to frictional losses. In all the investigated cases the generated entropy is clearly dominated by heat transfer irreversibility and attained a value of [0.94, 0.97]. Furthermore, increasing the Ri results in increasing the total entropy however, dominated by heat transfer, the increase is about 2%. The entropy generated peaks out between Ri 50 and 100, as can be seen at higher Ri number a slight decrease in Bejan number is noted, which is indicative of an increase in frictional entropy generation relative to entropy generated by heat transfer.

Shown in Figure 6 is the effect of undulations, N, 1–4 at a fixed ovoid angle of 0° and selected Ri of 1.

It appears that the undulation squeezes the space available for the global vortex Figure 6a, which enhances the convective heat transfer as shown by the increased perturbation of the isotherm plots in Figure 6b, the latter leads to the generation of a greater amount of entropy and the thermally driven irreversibility as seen in Figure 6c.

Shown in Figure 7a is the average heat transfer coefficient, which shows that increasing the undulation is directly proportional to increase Nusselt number.

At a low Ri number or in the mixed convection regime, the enhancement in heat transfer per undulation is nearly 9%, whilst in the free convection regime, at a higher Ri, it is approximately 10%. The entropy generation is dominated by heat transfer, as explained in Figure 7b. We also notice that the Bejan number peaks at between Ri 50 and 100. The effect of the applied magnetic field on the motion of the fluid and the associated convection is illustrated in Figure 8, having a Ri of 1 and ovoid cavity angle of 0°.

As the Hartman number is increased from 0 to 100, the Lorentz force suppresses the global cavity fluid motion and reduces the effect of the local cavity near the moving lid.

The magnetic field reduces convection and the conduction mode dominates. This is approved by the temperature contour distortions that are smoothed out with the increasing Ha number; this results in a stratified temperature field and an aligned temperature gradient between the two isothermal surfaces, as shown in Figure 8b. Naturally, this led to a decrease in the entropy generation attributed to heat transfer, as seen in Figure 8c, and entropy generation increased due to the magnetic field effect. Figure 9a shows the Nusselt number as the magnetic field is activated. The suppression of the convective effect results in the diminution of the heat transfer process and, hence, an extreme reduction in the Nusselt number.

Increasing the Hartman number from 25 to 100 has a marginal effect on the further decrease in the Nusselt number. Figure 9b shows the Bejan number increase with the increasing Hartman number, which can be attributed to irreversibility associated with magnetic field effects.

Generally, introducing nanoparticles in the cavity fluid causes it to flow slower, resulting in a more substantial temperature gradient and an increase in the heat transfer rate. Figure 10a shows the Nusselt number versus Ri for an increasing nanoparticles volume fraction.

Doubling the nanoparticle fraction increases the Nusselt number by approximately 10%. Figure 10b shows that increasing the volume fraction also increases the entropy generation, which can be attributed to an increase in the apparent viscosity of the fluid, the agent for fluid friction, and the source of irreversibility.

## 5. Conclusions

Numerical simulations of mixed convection within a concentric cavity with wavy boundaries are carried out. The inner cavity is an adiabatic oval, and the outer is a lid-driven square cavity. The space in the cavity is filled with a nanofluid. The convective heat transport is set up by the moving lid and heating from below. A uniform magnetic field is imposed on the working fluid. The dimensionless numbers of interest are Richardson, Hartman, and Bejan. The undulation of the heated wall and orientation of the inner cavity is varied. The results are displayed in in-stream function, isothermal, and isentropic contour plots.

Additionally, the Nusselt number and Bejan number are calculated. It is determined that a single incrementation of undulation increases the Nusselt number by an average of 9.5% within the investigated range (1 ≤ *N* ≤ 4). Furthermore, doubling the nanoparticle volume fraction increased the Nusselt number by nearly 8%. Meanwhile, incrementing the major ovoid axes from the horizontal plane by 10° raises the Nusselt number by 2.8% and 0.81% in the mixed convection regime (low Ri) and free convection regime (high Ri), respectively. The Nusselt number increases with increasing the Richardson number for any of the factors mentioned above, as the natural convection heat transfer mode dominates the mixed convection mode. An increase in entropy generation is associated with efficient thermal transport due to heat transfer irreversibility. Imposing a magnetic field shuts down the convection mode, rendering conduction heat transfer the sole mechanism of thermal transport.

## Figures and Tables

**Figure 1 nanomaterials-12-01392-f001:**
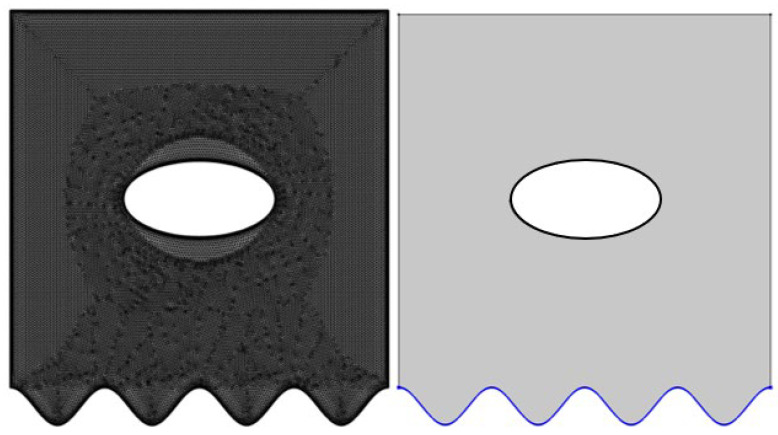
Physical model and mesh.

**Figure 2 nanomaterials-12-01392-f002:**
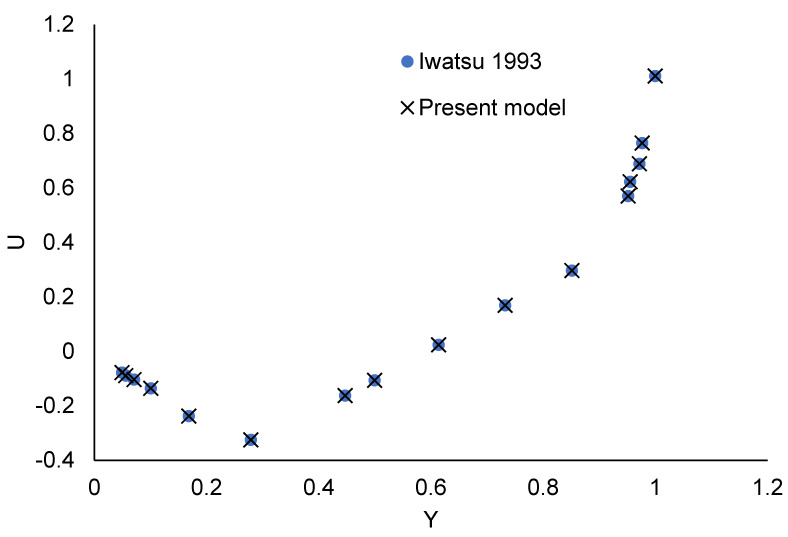
Validation of code velocity profile [56] Reproduced with permission from [56]. Elsevier, 1993.

**Figure 3 nanomaterials-12-01392-f003:**
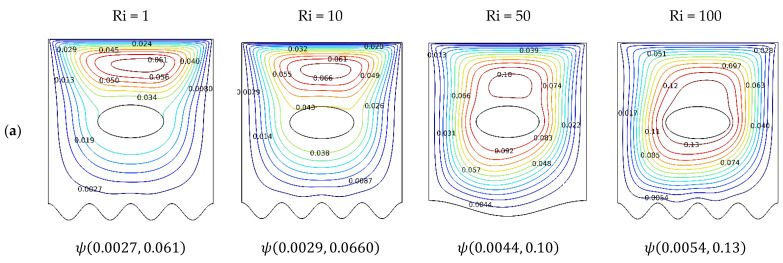
Streamlines, isotherms, and isentropic contours: γ=0; N=4; Ha=0; Ri:1, 10, 50, 100; φ=0.02. (**a**) ψ; (**b**) Θ; and (**c**) S.

**Figure 4 nanomaterials-12-01392-f004:**
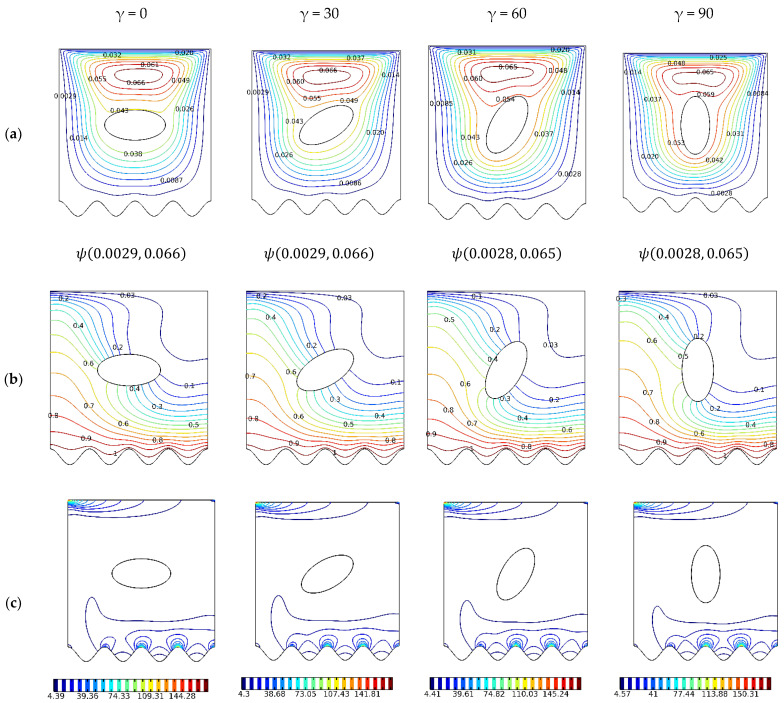
Streamlines, isotherms, and isentropic contours: $\gamma :\;0^{\circ },30^{\circ },60^{\circ },90^{\circ };\;N=4;\;Ha=0;\;Ri=1;\;\varphi =0.02$. (**a**) ψ; (**b**) Θ; and (**c**) S.

**Figure 5 nanomaterials-12-01392-f005:**
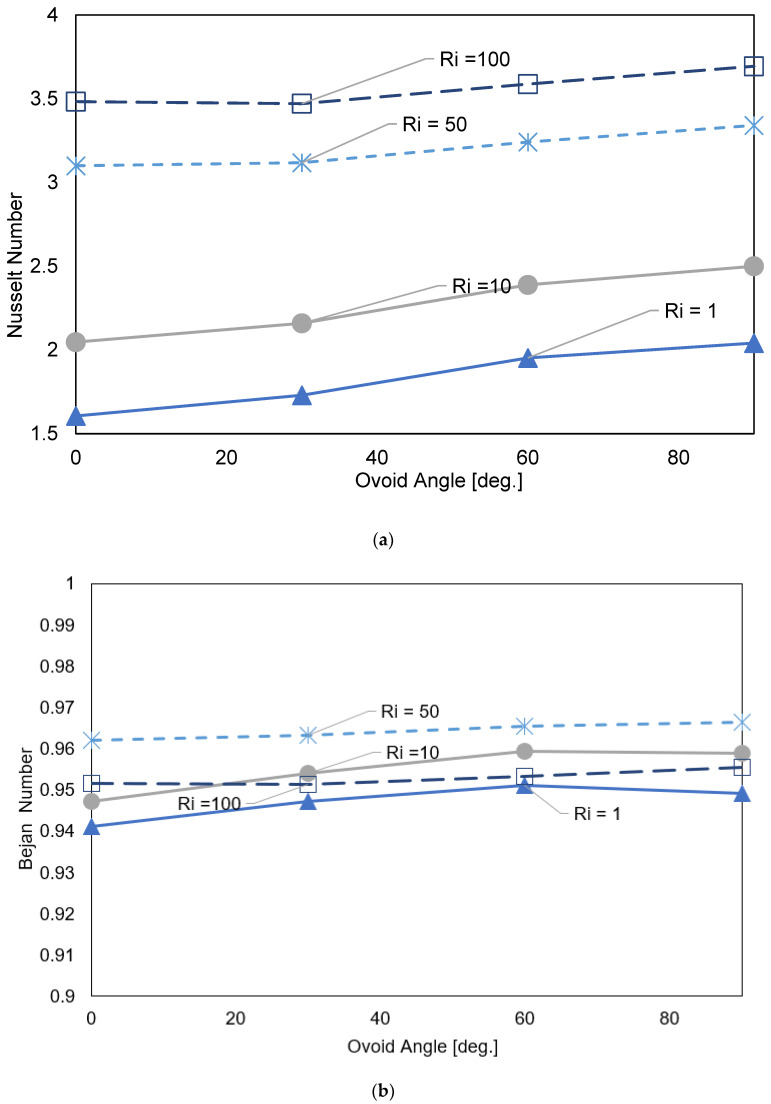
(**a**) Nusselt number: $\gamma :\;0^{\circ },30^{\circ },60^{\circ },90^{\circ };\;N=4;\;Ha=0;\;Ri=1,\;10,\;50,\;100;\;\varphi =0.02$. (**b**) $\mathrm{Bejan}\;\mathrm{number}:\;\gamma :\;0^{\circ },30^{\circ },60^{\circ },90^{\circ };\;N=4;\;Ha=0;Ri=1;\;\varphi =0.02$.

**Figure 6 nanomaterials-12-01392-f006:**
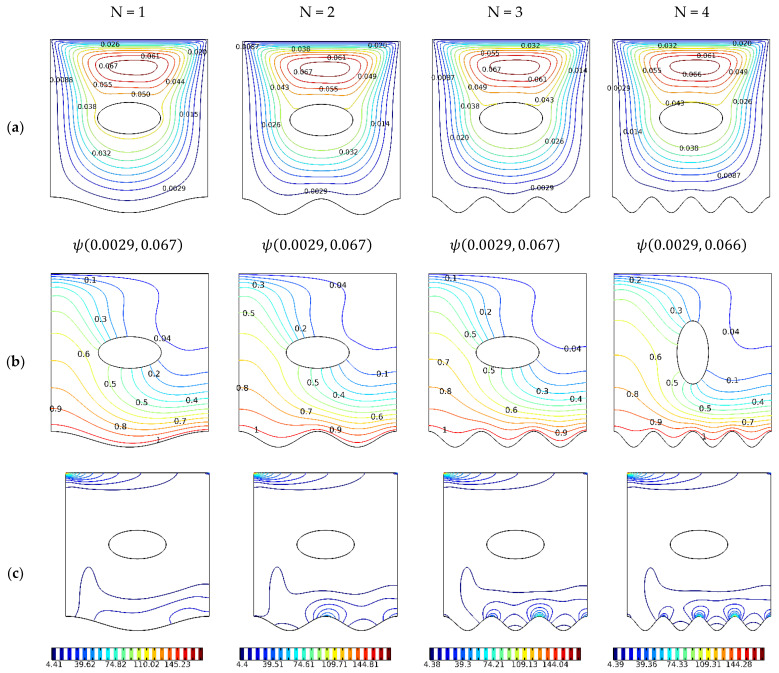
Streamlines, Isotherms, and Isentropic contours, $\gamma :\;0^{\circ };\;N=1,\;2,\;3,4;\;Ha=0;\;Ri=1;\;\varphi =0.02$. (**a**) ψ; (**b**) Θ; (**c**) S.

**Figure 7 nanomaterials-12-01392-f007:**
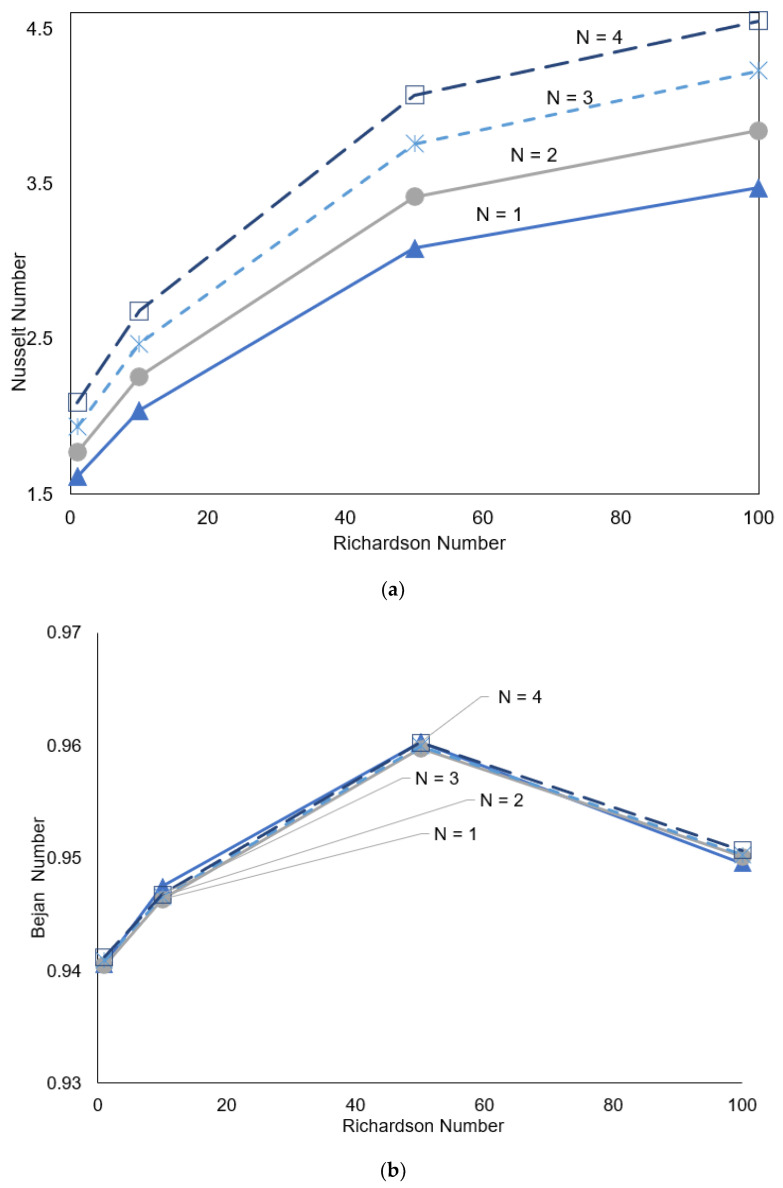
(**a**) Nusselt number, *γ*: 0°; *N* = 1, 2, 3, 4; *Ha* = 0; *Ri* = 1, 10, 50, 1; *φ* = 0.02. (**b**) Bejan number, *γ*: 0°; *N* = 1, 2, 3, 4; *Ha* = 0; *Ri* = 1; *φ* = 0.02.

**Figure 8 nanomaterials-12-01392-f008:**
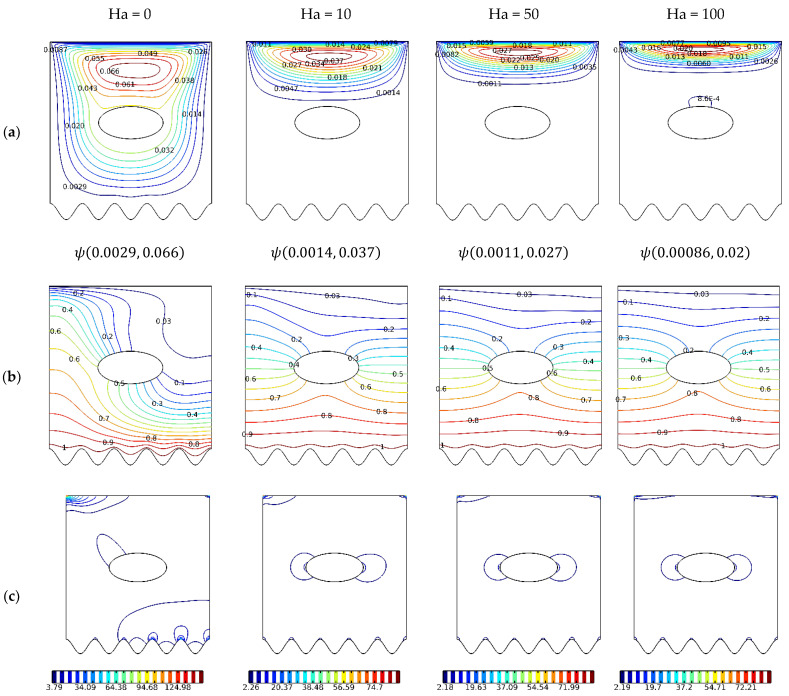
Streamlines, isotherms, and isentropic contours: *γ*: 0°; *N* = 4; *Ha* = 0, 25, 50, 100; *Ri* = 1; *φ* = 0.02. (**a**) ψ; (**b**) Θ; and (**c**) S.

**Figure 9 nanomaterials-12-01392-f009:**
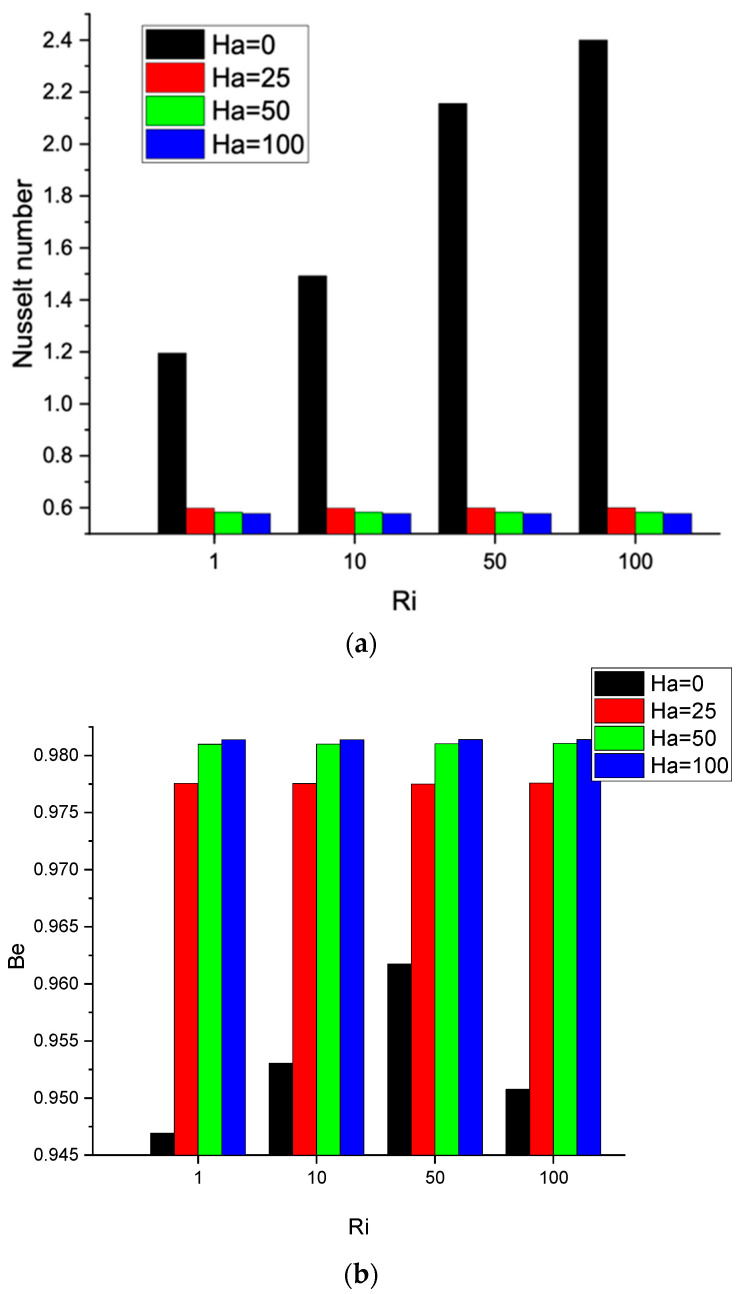
(**a**) Nusselt number: *γ*: 0°; *N* = 4; *Ha* = 0, 25, 50, 100; *Ri* = 1, 10, 50, 100; *φ* = 0.02; (**b**) Bejan number: *γ*: 0°; *N* = 4; *Ha* = 0, 25, 50, 100; *Ri* = 1, 10, 50, 100; *φ* = 0.02.

**Figure 10 nanomaterials-12-01392-f010:**
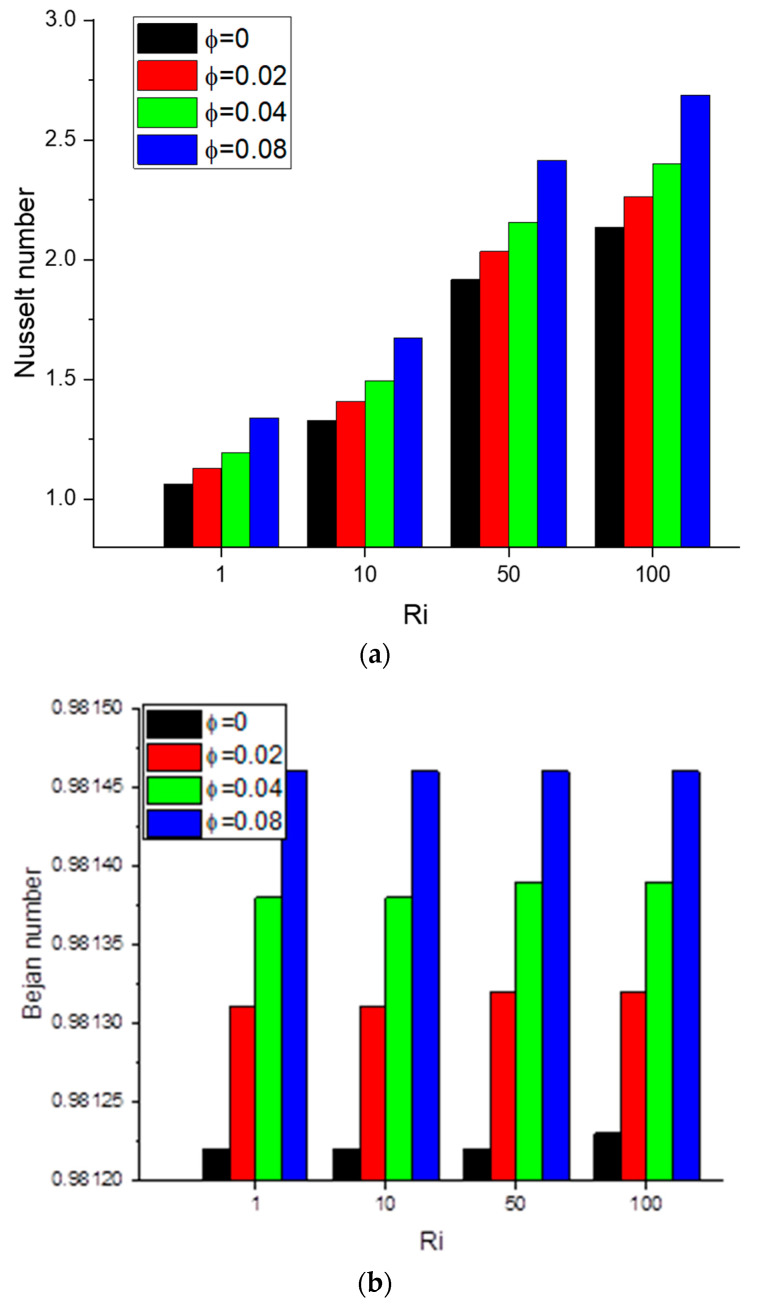
(**a**) Nusselt number: *γ*: 0°; *N* = 4; *Ha* = 0; *Ri* = 1, 10, 50, 100; *φ* = 0, 0.02, 0.04, 0.08; (**b**) Bejan number: *γ*: 0°; *N* = 4; *Ha* = 0; *Ri* = 1, 10, 50, 100; *φ* = 0.02, 0.04, 0.08.

**Table 1 nanomaterials-12-01392-t001:** Material properties at a temperature of 293 K. Reprinted/adapted with permission from Ref. [52]. Elsevier, 2017.

Material	$\rho \;\left[\mathrm{kg}/m^{3}\right]$	$C_{p}\;\left[J/\mathrm{kg}\cdot k\right]$	$\mu \times 10^{6}\;\left[\mathrm{Pa}\cdot s\right]$	$\beta \times 10^{5}\;\left[1/k\right]$	k [W/m·k]	σ [S/m]
Alumina (Al_2_O_3_)	3970	765	-	0.85	25	10^−10^
Water	997.1	4179	695	21	0.613	0.05

## Data Availability

Not applicable.

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
