# Peer review of "Hydrothermal and Entropy Investigation of Nanofluid Natural Convection in a Lid-Driven Cavity Concentric with an Elliptical Cavity with a Wavy Boundary Heated from Below"

_nanomaterials, 2022, doi:10.3390/nano12091392_

Round 1
Reviewer 1 Report
The contents are relevant to this journal. The main observations are listed below. I can recommend its publication but before publication, I suggest following revision.
- Authors should not be overused the well-known information in the abstract, even as background information. The abstract should be briefly written to describe the purpose of the research, the principal results, and major findings. Authors should revise it.
- In the Introduction, the literature review was not logically organized and all literatures cited seem separate descriptions without connections. The readers can’t know what the state-of-art methodologies or gaps the current study plans to resolve or fill, and how significant or what contribution the current study is?
- The authors should do a better job on commenting the results. A reasonable physical explanation should be provided for the observed trends, not only report what is graphically seen in the figures. More physical insight of the Discussion section is needed.
- Cite a reference to an equation, unless it is proposed here originally.
- Figures are of low quality which cause difficulty in reading. Improve the quality of figures.
- Authors may include the closely related work such as: Nanomaterials 2020, 10(3), 449; Symmetry, 2021, 13 (12), 2358; Nanomaterials 2021, 11(9), 2250; Journal of Thermal Analysis and Calorimetry, 140, 1121–1145 (2020); Nanomaterials 2022, 12(4), 663
- There are many typo-errors and language corrections requirements. At several places in the text the word spacing has not been taken care of. Typographic errors and word spacing can be fixed in the revised version for running a spell check.
- Validation part should be included.
- More computational details are required (i.e., methodology, convergence, validation, type of solver such as finite element or finite volume method, type and distribution of mesh, the utilized discretization scheme).
- There are various mistakes in writing and defining the symbols used in the manuscript. Please rectify these mistakes.
- Write the conclusion more precious.
Author Response
Responses to Reviewer: 1
- Authors should not be overused the well-known information in the abstract, even as background information. The abstract should be briefly written to describe the purpose of the research, the principal results, and major findings. Authors should revise it.
We would like to thank the reviewer for all constructive comments to improve the manuscript. The abstract has been modified.
- In the Introduction, the literature review was not logically organized and all literatures cited seem separate descriptions without connections. The readers can’t know what the state-of-art methodologies or gaps the current study plans to resolve or fill, and how significant or what contribution the current study is?
Done
- The authors should do a better job on commenting the results. A reasonable physical explanation should be provided for the observed trends, not only report what is graphically seen in the figures. More physical insight of the Discussion section is needed.
Done
- Cite a reference to an equation, unless it is proposed here originally.
Done,[29]
- Figures are of low quality which cause difficulty in reading. Improve the quality of figures.
Done
- Authors may include the closely related work such as: Nanomaterials 2020, 10(3), 449; Symmetry, 2021, 13 (12), 2358; Nanomaterials 2021, 11(9), 2250; Journal of Thermal Analysis and Calorimetry, 140, 1121–1145 (2020); Nanomaterials 2022, 12(4), 663
Done
- There are many typo-errors and language corrections requirements. At several places in the text the word spacing has not been taken care of. Typographic errors and word spacing can be fixed in the revised version for running a spell check.
Done
- Validation part should be included.
Please check fig 2a,b
- More computational details are required (i.e., methodology, convergence, validation, type of solver such as finite element or finite volume method, type and distribution of mesh, the utilized discretization scheme).
Please check Numerical Method and Validation section
- There are various mistakes in writing and defining the symbols used in the manuscript. Please rectify these mistakes.
Done
- Write the conclusion more precious.
Done
Reviewer 2 Report
An interesting and potentially valuable task is performed namely, numerical simulations of mixed-convection within a concentric cavity with wavy boundary. The key accent is on the effect of a magnetic field on the flow of nanofluid related to convective flow and temperature distribution due to heated from below wavy boundary. The results have a considerable application potential and are of interest to the readers of MDPI Nanomaterials. The manuscript could be accepted for publication following some technical improvements and additional clarifications.
- In order to be readable for a broader Nanomaterials’ audiences, some of the specific short terms should be explicitly defined, particularly when used for the first time, e.g. MHD, LDSC.
- The physical background for the introduction of the scaling parameters L for x and y, and U0 for u and v, should be at least shortly discussed and substantiated.
- The cited reference [ref**] in line 120 is unclear.
- On Fig.2a the data from Iwatsu et al. [49] and the present model are indistinguishable. Although a comparison is announced, there is no comment on this fact.
- The sentence “The momentum sink…..” (lines 259-264) is rather lengthy and unclear.
Author Response
Responses to Reviewer: 2
Comments and Suggestions for Authors
An interesting and potentially valuable task is performed namely, numerical simulations of mixed-convection within a concentric cavity with wavy boundary. The key accent is on the effect of a magnetic field on the flow of nanofluid related to convective flow and temperature distribution due to heated from below wavy boundary. The results have a considerable application potential and are of interest to the readers of MDPI Nanomaterials. The manuscript could be accepted for publication following some technical improvements and additional clarifications.
- In order to be readable for a broader Nanomaterials’ audiences, some of the specific short terms should be explicitly defined, particularly when used for the first time, e.g. MHD, LDSC.
The authors would like to thank the review for the constructive comments. The abbreviations are defined.
- The physical background for the introduction of the scaling parameters L for x and y, and U0 for u and v, should be at least shortly discussed and substantiated.
Done. “ Non-dimensionlization of the governing equation is carried out by scaling the equations using the characteristic length scale of the square cavity and the, L, and the velocity of the driven Lid, UO, as follows:
- The cited reference [ref**] in line 120 is unclear.
Done. [29]
- On Fig.2a the data from Iwatsu et al. [49] and the present model are indistinguishable. Although a comparison is announced, there is no comment on this fact.
The figure was changed to show the two data sets and a comment on the result is added “ perfect agreement between the two results is obtained.”
- The sentence “The momentum sink…..” (lines 259-264) is rather lengthy and unclear.
It is modified as follows: “The magnetic field reduces convection and conduction mode dominates. This is approved by the temperature contours distortions that are smoothed out with increasing Ha number, this results in a stratified temperature field and an aligned temperature gradient between the two isothermal surfaces as shown in Figure 8 (b)”
Reviewer 3 Report
The authors have examined mixed convection in a lid-driven cavity filled with nanofluid. The bottom wavy wall is heated while the upper lid is at a uniform cold temperature. The vertical walls are maintained in adiabatic conditions. The Richardson number (Ri-1-100) spans mixed convection and free convection regimes. The flow is subject to a varying magnetic field, Hartman number (Ha), 1-100. Nanofluid volume fraction, (φ), ranges between 0 and 0.08%. The effect of the various parameters on flow, thermal transport, and entropy generation is illustrated by the stream function, isotherms, and isentropic contours. In general, the manuscript is good, however, there are some important issues the authors must answer before, final acceptance.
- Novelty if this work is not clear. The authors must discuss it in the abstract and in the last paragraph of the introduction.
- Derivation of equations 2,3 and 4 must be included. The authors have added MHD and free convection terms in eq. 3 not in eq. 2 why? These calculi should be added in the appendix.
- Results of skin friction should be included.
- Viscous dissipation term is not included in the temp equation but is included in entropy generation?
- Nu and Be must be written in a normal form, not in integral form.
- Magnetic parameter big values are chosen, any physical reason?
Author Response
Responses to Reviewer: 3
Comments and Suggestions for Authors
The authors have examined mixed convection in a lid-driven cavity filled with nanofluid. The bottom wavy wall is heated while the upper lid is at a uniform cold temperature. The vertical walls are maintained in adiabatic conditions. The Richardson number (Ri-1-100) spans mixed convection and free convection regimes. The flow is subject to a varying magnetic field, Hartman number (Ha), 1-100. Nanofluid volume fraction, (φ), ranges between 0 and 0.08%. The effect of the various parameters on flow, thermal transport, and entropy generation is illustrated by the stream function, isotherms, and isentropic contours. In general, the manuscript is good, however, there are some important issues the authors must answer before, final acceptance.
- Novelty if this work is not clear. The authors must discuss it in the abstract and in the last paragraph of the introduction.
Done. The abstract and the last paragraph of the introduction has been modified
Abstract
“This work investigates mixed-convection in a lid-driven cavity. This cavity is filled with nanofluid and subjected to magnetic field. The concentric ovoid cavity orientation, (γ), 0o-90o and undulation number (N),1-4 are considered. The Richardson number (Ri) varies between 1-100. And nanofluid volume fraction, (φ), ranges 0 and 0.08%. The effect of the parameters on flow, thermal transport, and entropy generation is illustrated by the stream function, isotherms, and isentropic contours. Heat transfer is augmented at higher Ri, γ, N, φ. The simulations shows that the heat transfer is responsible for entropy generation while frictional and magnetic effects are marginal.”
Last Paragraph in introduction
“ In this study numerical simulations of mixed-convection within a concentric cavity with wavy boundary is conducted. The key accent is on the effect of a magnetic field on the flow of nanofluid, in a lid-driven in a square cavity, related to convective flow and temperature distribution due to heated from below wavy boundary. The governing equations are solved using the Galerkin Finite Element Method (GFEM) [7]. The acquired data are shown graphically using isotherms, streamlines, Bejan, and Nusselt values.“
- Derivation of equations 2,3 and 4 must be included. The authors have added MHD and free convection terms in eq. 3 not in eq. 2 why? These calculi should be added in the appendix.
The governing equations are referenced to [43]
[43] S. Jani, M. Mahmoodi, M. Amini, Magnetohydrodynamic free convection in a square cavity heated from below and cooled from other walls, Int. J. Mech. Aerosp. Ind. Mechatron. Eng. 7 (2013) 331–336.
In this work the cavity is filled with nanofluid, we consider the gravity vector is aligned with the y-axis and a uniform magnetic field is imposed along the x-axis.
- Results of skin friction should be included.
The focus of this work is on the heat transfer and also to high the effect of that and the magnetic field on the entropy generation
- Viscous dissipation term is not included in the temp equation but is included in entropy generation?
Equations are correct and compared with other references.
- Nu and Be must be written in a normal form, not in integral form.
Done
- Magnetic parameter big values are chosen, any physical reason?
Yes it is possible for physically, this is because the relative influence of Lorentz forces versus viscous forces is measured by the dimensionless Hartmann number Ha. At large Hartmann number, the flow can be analysed in terms of an inviscid core flow and viscous shear layers. In addition, strong magnetic fields are becoming widely available for research and industrial purposes, and understanding the asymptotic limit of large-Hartmann-number flows is becoming increasingly useful in metallurgy and other processes involving liquid metals, like liquid metal cooling or spallation neutron sources
Reviewer 4 Report
I recommend accepting the manuscript after the authors do the minor revisions included in the review report in the attachment file.

Author Response
Responses to Reviewer: 4
Comments and Suggestions for Authors
I recommend accepting the manuscript after the authors do the minor revisions included in the review report in the attachment file.
- Add more quantitative results to the abstract part.
The abstract was modified
“This work investigates mixed-convection in a lid-driven cavity. This cavity is filled with nanofluid and subjected to magnetic field. The concentric ovoid cavity orientation, (γ), 0o-90o and undulation number (N),1-4 are considered. The Richardson number (Ri) varies between 1-100. And nanofluid volume fraction, (φ), ranges 0 and 0.08%. The effect of the parameters on flow, thermal transport, and entropy generation is illustrated by the stream function, isotherms, and isentropic contours. Heat transfer is augmented at higher Ri, γ, N, φ. The simulations shows that the heat transfer is responsible for entropy generation while frictional and magnetic effects are marginal.”
- Would you please emphasize the novelty and difference of present work with previous works more?
The last paragraph in the introduction is modified
Last Paragraph in introduction
“ In this study numerical simulations of mixed-convection within a concentric cavity with wavy boundary is conducted. The key accent is on the effect of a magnetic field on the flow of nanofluid, in a lid-driven in a square cavity, related to convective flow and temperature distribution due to heated from below wavy boundary. The governing equations are solved using the Galerkin Finite Element Method (GFEM) [7]. The acquired data are shown graphically using isotherms, streamlines, Bejan, and Nusselt values.“
- All abbreviations must be defined as soon as they first appear in the manuscript.
Done.
- Please check all manuscript for typo and punctuation mistakes, such as Definite and Indefinite Articles (a, an, the), the capital letters in the middle of sentences, etc.
Done.
- It is recommended that the wordings and grammar of English should be rechecked throughout the present.
Done.
- There are some papers closely related to the study and therefore I suggest that the authors include them as much as possible as they are recent studies and within the scope of the manuscript as addressed by the authors.
- https://doi.org/10.1038/s41598-022-06134-6
- https://doi.org/10.1007/s13369-021-06015-6
Done.
- Add a nomenclature part to the manuscript.
Done.
- Would you please cite all equations that you did not derive? With these modifications, I think this already very good article can be improved somewhat and will be better. I then welcome it for publication after revision
Done
Round 2
Reviewer 1 Report
The changes made by the authors are in accordance with my recommendations given during the original review; therefore, I am in favor of its publication for this journal now.
Author Response
We would like to thank the reviewer for his constructive comments which improved the manuscript.
Reviewer 3 Report
The literature is full of similar problems, the authors have not included novelty even in the revision. Moreover, the convection terms are considered in the v-component of velocity, with no explanation why it is not considered along the u-component, therefore, the governing equations are not clearly modeled. If the authors can do detailed derivations of the equations, I will be happy to read the revision, otherwise, I am not feeling confident with the correctness of the present results, as velocity comes in entropy generation so those results are also doubtful, therefore, I do not recommend is for publication.
